# The Na⁺/H⁺ Exchanger NHX1 Controls H⁺ Accumulation in the Vacuole to Influence Sepal Color in *Hydrangea macrophylla*

Gaitian Zhang, Suxia Yuan, Hui Qi, Zhiyun Chu and Chun Liu *

Institute of Vegetables and Flowers, Chinese Academy of Agricultural Sciences, Beijing 100081, China
* Correspondence: liuchun@caas.cn; Tel.: +86-010-82109510

**Abstract:** *Hydrangea macrophylla* is popular for its unique physiological characteristics and changeable colors. Previous studies have shown that the pH of the vacuoles of the sepal cells of hydrangea affects the color of the sepals. Located on the vacuolar membrane, NHX1 is an important H⁺ proton pump that drives the exchange of metal ions. This proton pump affects the physiological environment by controlling the accumulation of H⁺ in the vacuole. In hydrangea, the *HmNHX1* gene has an open reading frame of 1626 bp and encodes a total of 541 amino acids. Bioinformatic analysis showed that *HmNHX1*, which encodes a Na⁺/H⁺ exchanger, is located on the vacuolar membrane. Tissue-specific expression analysis showed that the expression of this gene in the treatment group was higher than that in the control group. The ion flux in the vacuoles of colored hydrangea in the treatment group and the control group were measured, and the results showed that *HmNHX1* was indeed a Na⁺/H⁺ exchanger. When the results of the *HmNHX1* expression analysis and ion flux measurements are combined, it can be seen that *HmNHX1* regulates the accumulation of H⁺ in the vacuole, ultimately affecting the color of the plant.

**Keywords:** *Hydrangea macrophylla*; NHX1; flower color; ion flux measurements

## 1. Introduction

*Hydrangea macrophylla* is a unique ornamental plant species, and its main ornamental parts are the sepals of sterile flowers [1]. This species is native to East Asia [2]. Hydrangea is famous for its rich and variable colors, and the color of the sepals of some varieties can change to blue after aluminum application [3]. In vitro simulation experiments have shown that the aluminum ion content, the type of co-pigment, and the pH of the simulation solution affect the color of the sepals. The average pH of the vacuole of blue hydrangea plants is 4.1, while that of the vacuole of red plants is 3.3 [4,5]. In other species, the pH of the vacuole of purple-blue flowers is higher than that of red flowers [6–8]. This occurs because when the pH is higher, the absorbance of the chromogenic substance, i.e., co-pigmentation of anthocyanins and co-pigments, is bathochromically shifted [9,10].

Studies have shown that the vacuolar pH in relation to plant flower color is mainly regulated by two different types of proton pumps that rely on a H⁺ concentration gradient to transport H⁺: P-type ATPase and Na⁺(K⁺)/H⁺ exchangers [6,11]. The Na⁺(K⁺)/H⁺ exchanger is a CPA (monovalent cation reversal protein) that is involved in the regulation of the cell cycle and proliferation, salt tolerance, vesicle trafficking, and biogenesis [12]. In most related studies, *NHX1* has been found to encode a sodium–hydrogen antiporter involved in salt tolerance; its main role is to transport Na⁺(K⁺) into the vacuole and, at the same time, replace H⁺. This method not only ensures the stability of the solution environment inside the vacuole but also effectively isolates the metal ions inside the vacuole to maintain the normal biological activities of cells [13–15]. However, this also increases the pH of the vacuole.

The color of *Ipomoea tricolor* changes from pink to purple or blue from the bud stage to the opening stage. The pigments do not change; the vacuole's pH continues to increase.

Related research has pointed out that the key gene that causes the blue corollas of Japanese morning glory is NHX1 [16]. LnNHX1 was also the first protein identified to regulate the increase in pH of the vacuole and cause the flower color to become blue [6]. Obviously, NHX1 is related to changes in flower color. Therefore, this study took *HmNHX1* as the research object to explore the role of NHX1 in the color change of hydrangea sepals.

## 2. Materials and Methods

### 2.1. Plant Materials and Sequencing

This experiment used two-year-old 'Bailmer' cuttings as test materials (only the sepals were used). All plants were grown in the greenhouse of the Institute of Vegetables and Flowers, Chinese Academy of Agricultural Sciences, and all pots were 15 cm in size. Huaduoduo compound fertilizer was regularly provided as the experimental fertilizer. Aluminum was supplied to the control, starting when the hydrangea internodes were no longer elongated, terminal buds were evident, and plant height was fixed. Aluminum (500 mL of 6 g/L $Al_2(SO_4)_3 \cdot 18H_2O$, pH = 4.5) was applied once a week after the terminal buds had formed. Eight pots (with three pots constituting one biological replication) were included in the aluminum treatment group and in the control group. Sampling was carried out with a mixture of three different plants and performed at three different stages (the bud stage (S1), the coloring period (S2), and the blooming period (S3)) (Figure 1). Each material had three biological replicates.

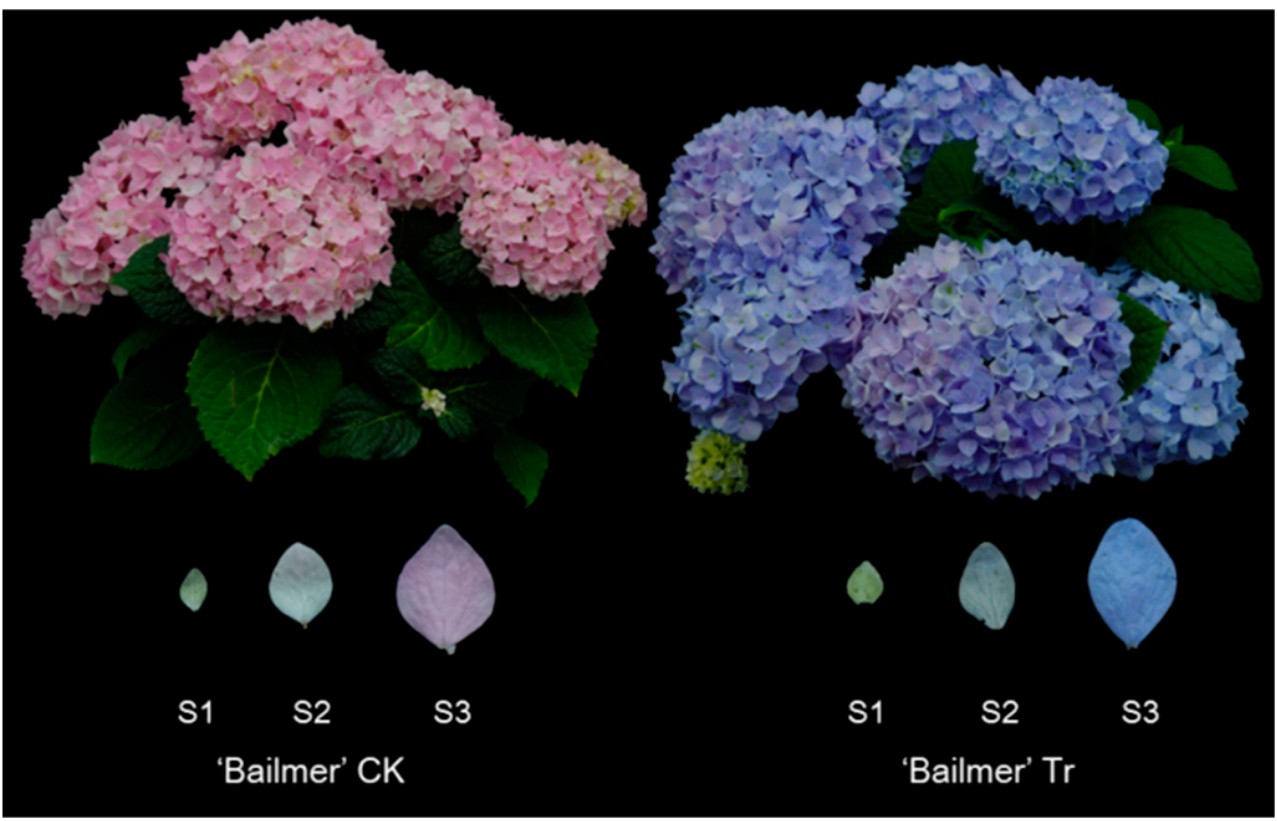

**Figure 1.** 'Bailmer' materials used in this study. CK, control group; Tr, treatment group; S, stage.

### 2.2. RNA Extraction and cDNA Synthesis

RNA was extracted with an EasySpin Plus Plant RNA Kit (Aidlab, Beijing, China). The RNA concentration was measured by an ultraviolet-visible spectrophotometer (Miulab, ND-100, Hangzhou, China). The RNA was reverse-transcribed with a PrimeScript RT Reagent Kit with gDNA Eraser (Takara, Dalian, China). The reaction volume for reverse transcription was 20 μL. Each reaction system included 500 ng of RNA.

### 2.3. Full-Length Amplification of the HmNHX1 Coding DNA Sequence (CDS)

The sequence of *ItNHX1* (*Ipomoea tricolor* NHX1, AB292774) [17] was queried (via BLAST), and the sequence was obtained (https://www.ncbi.nlm.nih.gov) (accessed on 17 September 2020). We used the sequence of *ItNHX1* to search for *HMNHX1* in the single-molecule real-time (SMRT) sequencing results.

We used Bailmer plants (at S3) as the experimental materials. According to the isoform data of *HmNHX1* screened by single-molecule real-time sequencing, the obtained sequence was used to predict the open reading frame (ORF) via NCBI ORFfinder (https://www.ncbi.nlm.nih.gov/orffinder) (accessed on 14 January 2021), and the results were inputted into the NCBI database to compare the length of the CDS of *HmNHX1* with that of the homologs in other species. Primers containing the *HmNHX1* ORF were designed (*HmNHX1*-F, ACATGTGAT-GTGATGCTTAGTTCGGAAG; *HmNHX1*-R, GACCAACAAGTGGGCGACAATCTGTAT) via Integrated DNA Technologies (https://sg.idtdna.com/calc/analyzer) (accessed on 16 January 2021). A KAPA HiFi HotStart ReadyMix PCR Kit (Roche, Indianapolis, IN, USA) was then used to amplify the target fragment (the reaction system consisted of 25 μL of $2\times$ KAPA HiFi HotStart ReadyMix, 3 μL of 5 μM forward primer, 3 μL of 5 μM reverse primer, 3 μL of cDNA template, and 16 μL of sterilized ddH$_2$O). The PCR program was as follows: 95 °C predenaturation for 3 min; 35 cycles of 98 °C denaturation for 20 s, 60 °C annealing for 15 s, and a 72 °C extension for 1.5 min; and a 72 °C final extension for 1.5 min. A Bio-Rad T-100 PCR thermal cycler instrument was used. The PCR products were subsequently detected via 1% (*w/v*) agarose gel electrophoresis.

### 2.4. Gel Extraction

The target bands were removed and placed into a preweighed 2 mL centrifuge tube. The weight of the gel was approximately 400 mg. An EasyPure Quick Gel Extraction Kit (TransGen, Beijing, China) was used to recover the gels.

A Zero Background pTOPO-Blunt Simple Cloning Kit (Aidlab, Beijing, China) and DH5$\alpha$ chemically competent cells (Tsingke, Beijing, China) were used for transformation. Luria–Bertani medium was used for the colony culture (100 mg/L ampicillin). We used the bacterial liquid as the amplification template to perform PCR amplification and sent the bacterial liquid corresponding to the correct amplification length for sequencing (Sangon Biotech, Shanghai, China).

### 2.5. Bioinformatic Analysis

The sequence of *HmNHX1* was predicted via the NCBI ORFfinder (https://www.ncbi.nlm.nih.gov/orffinder) (accessed on 2 February 2021) program. Cell-PLoc 2.0 (http://www.csbio.sjtu.edu.cn/bioinf/Cell-PLoc-2/) (accessed on 2 February 2021) was used to predict the subcellular location of the protein encoded by the *HmNHX1* gene. SOPMA (https://npsa-prabi.ibcp.fr/cgi-bin/npsa_automat.pl?page=npsa_sopma.html) (accessed on 2 February 2021)was then used to predict the secondary structure of the protein encoded by *HmNHX1*. SMART (http://smart.embl.de/) was used to predict the conserved regions and basic functions of the protein encoded by *HmNHX1*. ClustalX and ESPript (https://espript.ibcp.fr/ESPript/cgi-bin/ESPript.cgi) (accessed on 2 February 2021) were used for sequence alignment. MEGA 7 was subsequently used to construct a phylogenetic tree. MEME (https://meme-suite.org/) (accessed on 23 February 2022) was used to analyze the motif of HmNHX1. Figures were drawn with Origin95 and Inkscape.

### 2.6. Tissue-Specific Expression Analysis of HmNHX1

The standard curve was used to calculate the copy number of *HmNHX1* in each sample. The concentration of the plasmids returned by Sangon Biotech was 223.3 ng/μL, which was used as a benchmark for different dilution factors ($10^0$, $10^{-1}$, $10^{-2}$, $10^{-3}$, $10^{-4}$, $10^{-5}$, $10^{-6}$, $10^{-7}$, $10^{-8}$, and $10^{-9}$). We chose the sample data with dilutions of $10^{-5}$, $10^{-6}$, $10^{-7}$, $10^{-8}$, and $10^{-9}$ to construct the standard curve because their Cq values were similar. Plasmid copy numbers were calculated by the plasmid copy number calculation formula. The

logarithmic value of the plasmid copy number for different concentrations of plasmids was used as the abscissa, and the Cq value of the corresponding plasmid concentration was used as the ordinate (we included the Cq values of each sample). The instrument used for real-time fluorescence quantification was a Light Cycler 480 II (Roche, Basel, Switzerland), and the fluorescence was quantified with Forget-Me-Not qPCR Master Mix (Biotium, Fremont, CA, USA). The reaction volume was 10 μL, which consisted of 1 μL of the cDNA template (25 ng of RNA substrate template in each reaction), 0.5 μL of the amplification primers (5 μM), 5 μL of 2× Forget-Me-Not qPCR Master Mix, and 3 μL of sterile dH$_2$O. The PCR program was as follows: enzyme activation at 95 °C for 2 min, denaturation at 95 °C for 5 s, annealing at 60 °C for 10 s, and extension at 72 °C for 20 s. The sequences of the primers used for fluorescence quantification included TGATGCCA-CATCAGTTGTGCTG (forward) and CACCCAGCAAAGTGCTTGTGAG (reverse). All materials had three experimental replicates.

### 2.7. Ion Flux Measurements of Bailmer Vacuoles

Sepal protoplast isolation was performed according to the method of Yoshida et al. [4], with modifications. We prepared 0.008% poly-L-lysine (*w/v*, FW: 150,000–300,000, Sigma, St. Louis, MO, USA) [18], and then smeared onto disposable petri dishes, which were subsequently placed in a refrigerator to dry for later use. A noninvasive microtest technology (NMT-YG-100, Younger USA, LLC., Amherst, MA, USA, 01002) was used to measure the H$^+$, Na$^+$, and K$^+$ fluxes of the vacuoles of pink and blue sepals of Bailmer plants.

## 3. Results

### 3.1. HmNHX1 Bioinformatic Analysis

The results were analyzed, after which the primers were designed and amplified. The amplified product was sequenced, and its full length was 1861 bp. *ItNHX1* was the first gene identified to be associated with flower color and pH changes. The amino acid sequence similarity between lnNHX1 and *HmNHX1* was 78.29%.

ORFfinder predicted that the *HmNHX1* CDS has a total length of 1626 bp and encodes 541 amino acids. The protein encoded by *HmNHX1* is located on the vacuolar membrane, and its secondary structure consists of 44.92% α-helices, 31.79% random coils, 18.85% extended chains, and 4.44% β-turns. This protein functions as a Na$^+$/H$^+$ exchanger. The conserved functional region is formed by amino acids located from 24 to 444.

*HmNHX1*-related sequences in the NCBI database were queried, after which multiple sequence alignment and phylogenetic tree analysis were performed. We used the sequences of 17 species to construct evolutionary trees. These species included woody plant species and herbaceous plant species (gramineous plant species, model plant species, etc.). From the results of the phylogenetic tree analysis, *Hydrangea macrophylla*, *Camellia sinensis*, *Vitis vinifera*, *Hibiscus syriacus*, and *Olea europaea* clustered on the same three-level branches; these results contrast with those of herbaceous plant species, for which the clustering was obviously more diversified and had high similarity in the conserved structure interval. However, Motifs 1–9 showed that NHX1 of all species had high similarity in the conserved structure interval (Figure 2).

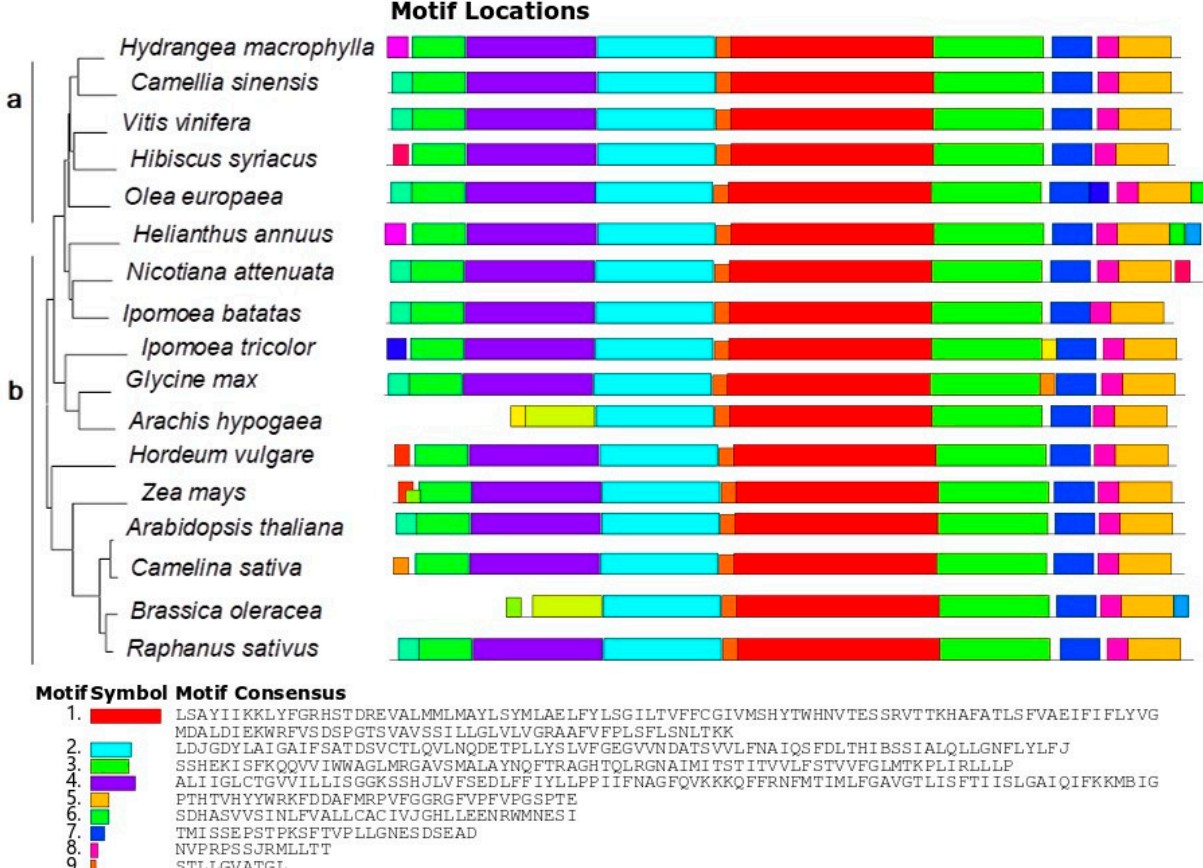

**Figure 2.** Phylogenetic tree of NHX1. Part (**a**) includes woody plants, and Part (**b**) includes herbaceous plants. The figure on the right shows the analysis of the NHX1motif for the corresponding species: *Hydrangea macrophylla*, *Camellia sinensis* (XM_028205587.1), *Vitis vinifera* (NM_001280886.1), *Hibiscus syriacus* (XP_039020832.1), *Olea europaea* (XM_022986696.1), *Helianthus annuus* (XM_022146020.2), *Nicotiana attenuata* (XM_019398501.1), *Ipomoea batatas* (AFQ00709.1), *Ipomoea tricolor* (AB292774.1), *Glycine max* (NM_001250237.2), *Arachis hypogaea* (XP_025680083.1), *Hordeum vulgare* (ANS57040.1), *Zea mays* (AAP20428.1), *Arabidopsis thaliana* (NM_122597.3), *Camelina sativa* (XP_010455152.1), *Brassica oleracea* (XP_013611175.1), and *Raphanus sativus* (XP_018440589.1).

### 3.2. Expression Analysis of HmNHX1

A standard curve was constructed to analyze the expression of *HmNHX1* in Bailmer hydrangea sepals (Figure 3a). The melt curve showed that the primer was well specified (Figure 3b). The expression patterns in the control and treatment groups showed that the expression of *HmNHX1* at S1 and S2 was significantly higher than that in the control group. The expression of *HmNHX1* at S3 was higher than that in the control group, although the results were not significant. In general, the expression level of *HmNHX1* in the treatment group was higher than that in the control group in the same period. The expression of *HmNHX1* gradually increased with plant growth in both the treatment and control groups. These results suggest that the amount of $H^+$ in the vacuole required to maintain the stability of the blue chromogenic substance is less than that required for the pink chromogenic substance (Figure 3c).

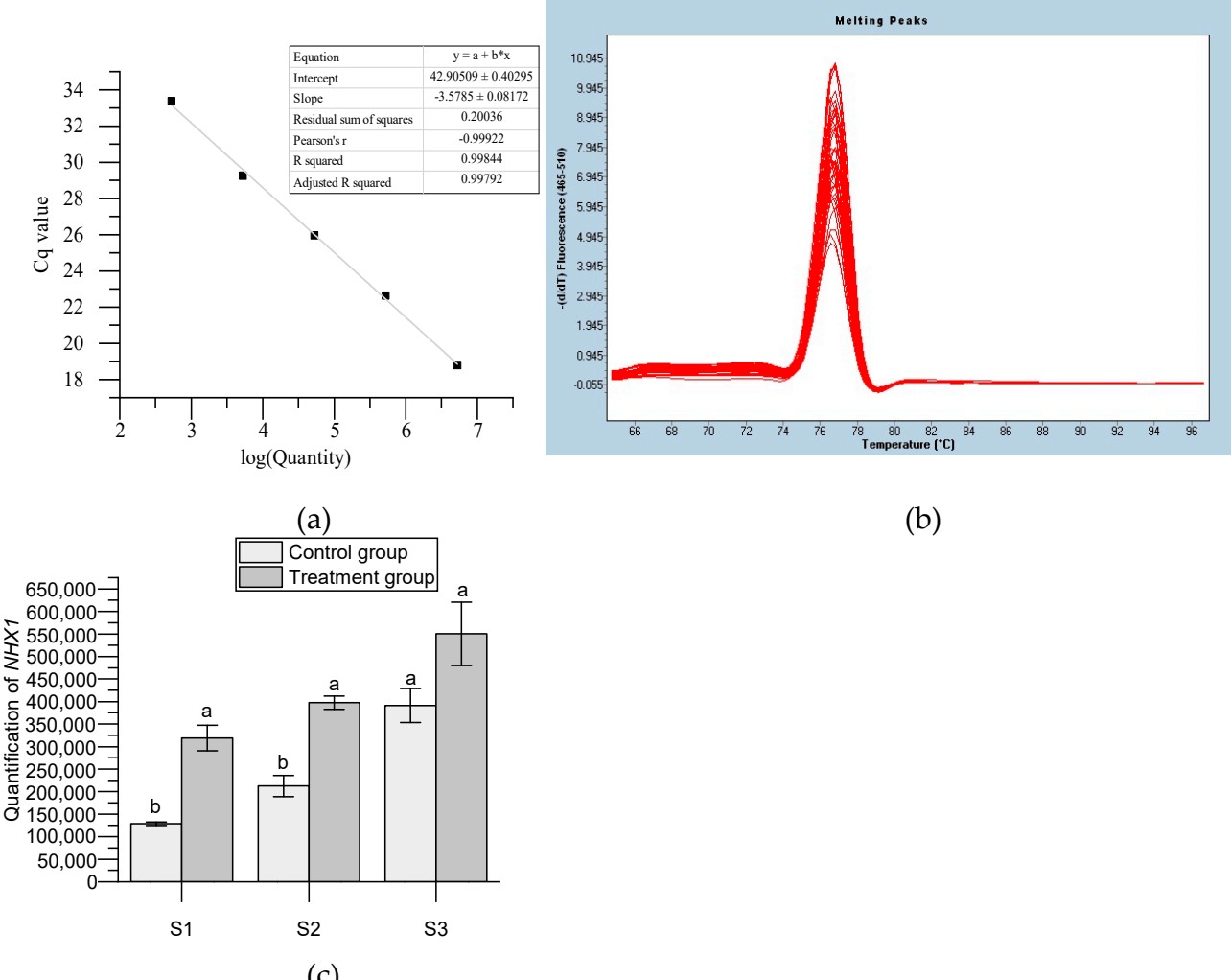

(a)                                                                                               (b)

(c)

**Figure 3.** Quantification of the expression of *HmNHX1*. (**a**) Standard curve of quantity. The range of concentration with stable fluorescence values and good linear ratios was selected. The logarithm of the copy number was used as the $x-$axis, and the number of reaction cycles (Cq) was used as the $y-$axis. (**b**) Melting curve of *HmNHX1*. (**c**) Expression of *HmNHX1* at different developmental stages of sepals under two different treatments in Bailmer hydrangeas ($p < 0.05$).

### 3.3. Relationship between NHX1 and Ion Flux Measurements

Protoplasts were isolated from pink and blue sepals of Bailmer hydrangeas at the full blooming stage, and measurements of the $H^+$, $K^+$, and $Na^+$ currents were performed. The results of the ion flux measurements showed that the vacuolar $H^+$ of blue sepals tended to out the vacuole, $K^+$ tended to exit, and $Na^+$ tended to enter, whereas the vacuolar $H^+$ of pink sepals tended to enter the vacuole, $K^+$ tended to enter, and $Na^+$ tended to exit. Combining these results with those of the analysis of the *HmNHX1* expression patterns, we inferred that the amount of $H^+$ that accumulated in the vacuoles of pink sepals was higher than the amount of $H^+$ that accumulated in the vacuoles of blue sepals, and that NHX1 is a $Na^+/H^+$ exchange pump, not a $K^+/H^+$ exchange pump (Figure 4).

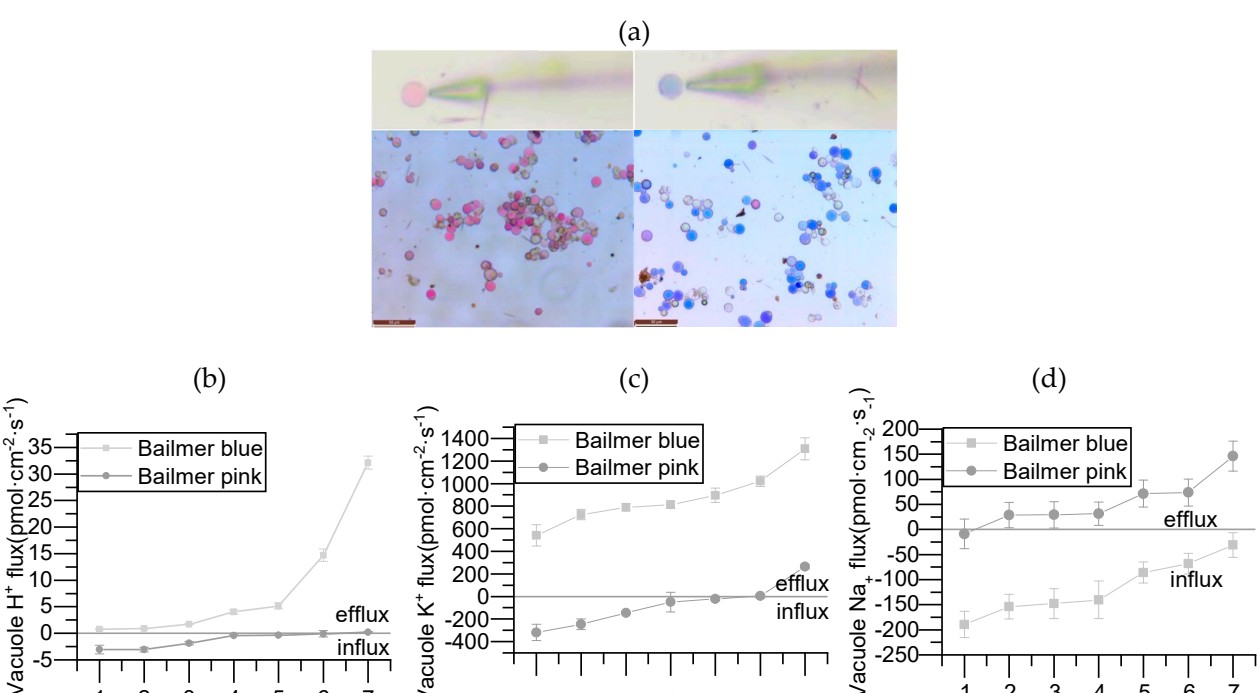

**Figure 4.** Protoplast and ion flux in Bailmer hydrangea. The ion flux of 7 vacuoles was measured in each pink and blue sepal. (**a**) Protoplasts and the corresponding vacuole ion flux measurements (bar = 50 μm). (**b**) H⁺ flux measurements. (**c**) K⁺ flux measurements. (**d**) Na⁺ flux measurements. Positive values show that ions flowed out of the vacuoles, and negative values show that ions flowed into the vacuoles.

## 4. Discussion

For most plants, flower color serves as a visual signal to attract insects to spread pollen and promote reproduction [19]. Anthocyanins, which are secondary metabolites, are the key substances for flower color formation and are stored in the vacuoles of plants [20,21]. Research related to blue flower formation in hydrangeas is continuing, and key genes regulating the DFR pathway have been identified [22]. However, the color of flowers is not only based on anthocyanins. The pH of the vacuole also is a key factor in maintaining the stability of anthocyanins in the vacuole, and the pH has an important influence on the color of plants' floral organs [23]. In in vitro simulation experiments, it was found that the absorbance of a solution changed when the pH changed, even when the amount of pigments was the same [24]. In the process of ion flux measurements, we found that the H⁺ of the vacuoles of blue sepals had an obvious efflux trend compared with that of the vacuoles of pink sepals, suggesting that the pH of the vacuoles of blue sepals was higher than that of the vacuoles of pink sepals. These results were consistent with those of previous studies [4].

In contrast to H⁺-ATPase and H⁺-PPase, which uses the H⁺ concentration gradient to pump H⁺ outside of the vacuoles and generate energy, NHX1 consumes energy and pumps H⁺ to the outside of the vacuoles. The Na⁺/H⁺ antiporter is crucial for the regulation of cellular salt and pH [25]. In the process of studying *Ipomoea tricolor*, researchers found that *ItNHX1* was the most important regulatory gene that controls the increase in pH in the vacuole of tricolor morning glory and the formation of the blue color of the corolla [18]. After quantitative analysis of *HmNHX1*, it was found that the expression trend of *HmNHX1* was consistent with the phenotype of the change in flower color; that is, at the same time point, the expression level of *HmNHX1* was higher in the vacuoles of blue sepals than in those of pink ones. Experiments have shown that *Ipomoea tricolor NHX1* encodes a K⁺/H⁺ exchanger, while *InNHX1* encodes a Na⁺/H⁺ exchanger; nonetheless, the homology between these two exchangers was as high as 92.7% [18,26].

The *ItNHX1* sequence was used for sequence alignment together with transcriptome data from *Hydrangea*. The results of the monoclonal sequencing of the obtained sequence were subjected to bioinformatic analysis, and the results showed that *HmNHX1* in Hydrangea species encodes a unique eukaryotic $Na^+/H^+$ exchanger. To further validate the results of bioinformatic analysis, we performed ion flux measurements. If the protoplasm was still present in the cell wall, we could not easily judge whether the probe was accurately close to the vacuole or other organelles, so we isolated the protoplasm to facilitate our detection of vacuolar ion fluxes. The noninvasive microtest technology can measure the flow rate and concentration of various ions over a period of time while maintaining the activity of the cells [27]. Ion flux was closely related to various cellular life activities, and many life activities are altered differently by the ion flux. This assisted in verifying the functionality of some ion pumps [27]. After measurements and identification of the ion fluxes, according to the existing results of *NHX1*-related research [28] and in combination with the results of the quantitative expression analysis, it was proven that NHX1 in hydrangea is a $Na^+/H^+$ exchanger and is related to color changes.

The main anthocyanin component of hydrangea is delphinidin. In in vitro simulation experiments with hydrangea, co-pigments in the simulated vacuolar solution were determined to be 5-caffeoylquinic acid (5cq) or 5-p-coumarinic acid (5pcq); moreover, when the vacuolar solution included enough $Al^{3+}$ and when the pH was approximately 4, the simulated solution appeared blue [5]. In other words, the complexes of delphinidin and 5cq (5pcq) and $Al^{3+}$ together produced a blue color in the solution at pH 4. Indeed, related studies have shown that this process of $Al^{3+}$ absorption in hydrangea, which results in the sepals turning blue, is a way for the plants to cope with aluminum stress and alleviate the effects of aluminum toxicity. Similar to the in vitro simulation experiments, hydrangea transports and isolates $Al^{3+}$ within the vacuoles to avoid aluminum stress. $Al^{3+}$, together with delphinidin and the co-pigments, formed a chromogenic substance capable of producing a purple-blue color [29,30]. According to the results of the present experiment, when the Bailmer plants were stressed with aluminum, the main role of *HmNHX1* may have been to adjust the concentration of $H^+$ in the vacuole, affecting the vacuolar solution content and maintaining homeostasis of the vacuole. Then, because of changes in the solution content inside the vacuole, the material state of the chromogenic substance of hydrangea underwent some degree of change, which established conditions to ensure that the hydrangea formed a blue chromophore, ultimately leading to phenotypic changes in the hydrangea sepals. There have been many analytical chemistry-related reports on color changes in hydrangea, and the related components and formation processes have been thoroughly elucidated via in vitro simulation experiments. However, in terms of the genes related to color regulation in hydrangea sepals and their functional verification, further investigations and research are needed.

## 5. Conclusions

This study showed that the blue sepal formation mechanism of hydrangea is not exactly the same as the mechanism of the change in the corolla color from purple to blue in Japanese morning glory. The key to blue color formation in hydrangea sepals is that after the hydrangea has absorbed a certain amount of aluminum ions, the plant responds to a series of biological reactions that may cause poisoning, and this reaction lays the physiological and biochemical foundation for the hydrangea's sepals to turn blue. *NHX1* is one of the genes involved in this biological regulation and is mainly responsible for $Na^+/H^+$ replacement in the vacuoles, which affects the hydration inside the vacuole of the hydrangea sepal and lays the foundation for the formation of blue hydrangea flowers.

**Author Contributions:** Conceptualization, C.L., S.Y. and G.Z.; methodology, C.L., S.Y. and G.Z.; software, G.Z.; investigation, G.Z., H.Q. and Z.C.; resources, C.L.; data curation, G.Z.; writing—original draft preparation, G.Z.; writing—review and editing, C.L.; supervision, C.L.; project administration, C.L.; funding acquisition, C.L. All authors have read and agreed to the published version of the manuscript.

**Funding:** This research was funded by the Central Public Interest Scientific Institution Basal Research Fund (IVF-BRF2020021) and the Science and Technology Innovation Program of the Chinese Academy of Agricultural Science (CAAS-ASTIP-2020-IVFCAAS).

**Institutional Review Board Statement:** This study does not involve humans or animals.

**Informed Consent Statement:** Not applicable.

**Data Availability Statement:** The data that support the findings of this study have been deposited into the CNGB Sequence Archive (CNSA) of the China National GeneBank database (CNGBdb) and the NCBI Sequence Read Archive (SRA) of the National Center for Biotechnology Information. The mRNA sequence data have been submitted to the NCBI GenBank database. Because of data confidentiality issues, public inquiries will be available after 10 May 2024.

**Acknowledgments:** We are in gratitude to the National Flower Improvement Center and Laboratory of Horticultural Crop Biology and Germplasm Creation for providing the facilities.

**Conflicts of Interest:** The authors declare no conflict of interest.

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
