# Peer review of "The Na+/H+ Exchanger NHX1 Controls H+ Accumulation in the Vacuole to Influence Sepal Color in Hydrangea macrophylla"

_2037-0164, doi:10.3390/ijpb14010022_

Round 1
Reviewer 1 Report
This is an interesting paper that many people will want to read. However, more information is needed in the introduction to support the methods you are using. For example, why do you isolate sepal protoplasts? What is the NMT technology used to measure vacuole fluxes? What even are vacuole fluxes? Many readers will not be familiar with these terms and methods, so please introduce them in order to create confidence in your experiment.
26: Italicize Hydrangea macrophylla
34: Define chromogenic.
56: How were the plants maintained? Were they unrooted or rooted cuttings? Did you apply the Al solution to media? What size were the pots? What kind of growing media was used?
65: RNA was extracted from where? Sepals? Or the whole plant?
65-80: This section is confusing because you describe RNA extraction twice. Please separate these by topic. One should be “RNA sequencing” and the other should be “Expression Analysis”.
70: Did you use whole-plant RNA or sepal RNA?
70: Single quote around ‘Bailmer’
73 – 75: It seems like you used SMRT sequencing, but the details are not provided. Please rewrite these Methods so that RNA extraction, RNA sequencing, and RNA sequence analysis are clearly tied together.
95: Please provide the details for the PCR amplification of bacterial liquid
99: This work would be better if you could verify your sequence against a reference genome. Why not used the genome published by Nashima et al? Or contact Lisa.Alexander@usda.gov for a fully annotated bigleaf hydrangea genome.
111: More details regarding the plasmids is needed.
128 – 132: This should be in the Materials and Methods. How did you sequence ItNHX1?
134: Please use the reference genome for hydrangea to check this.
Please italicize species names in this section and throughout.
Figure 3 – These figures are too small to see. Please relabel them as separate figures so they are large enough to view.
267: Define antiporter
296: Define chromophore
Author Response
请参阅附件。

Reviewer 2 Report
Here are my specific comments:
- Check the manuscript and correct all scientific names
Abstract: Give full scientific name for Hydrangea.
- The scientific names are not in italics
- HmNHX1 gene is indicated differently- correct it
Introduction:
- Hydrangea macrophylla - italics
-In vitro - italics - check throughout the manuscript
-Is it NHX1 or HmNHX1 ? - be specific
-line 47 - Ipomea tricolor - I did not clearly understand why authors mentioned about this here.
Materials and methods:
- 2.1 - Plant materials and sequencing:- Please modify the paragraph two and three in this.
- This can also be separated as (for example) 2.1 - Plant materials
2.2 - Sequencing
- Please correct the scientific names and also the names of genes and proteins.
-Figure 4 - Pls. check the caption - "Positive values showed ion outflowed vacuoles, and negative values showed ion outflowed vacuoles'. Are they the same or different ?
-'pH refers to the hydrogen ion concentration, meaning the ratio of the number of hydrogen ions in a solution to the total number of substances'. - I don't think that it is necessary to mention this in the discussion part.
